The influence of pharmacologically-induced affective states on attention bias in sheep

http://orcid.org/0000-0002-4571-2285 Monk Jessica E. 1 2 3 jessica.monk@csiro.au
Lee Caroline 1 caroline.lee@csiro.au
Belson Sue 1
http://orcid.org/0000-0001-9497-5148 Colditz Ian G. 1
Campbell Dana L.M. 1
1 Agriculture and Food, CSIRO , Armidale, NSW , Australia
2 School of Environmental and Rural Science, University of New England , Armidale, NSW , Australia
3 Sheep CRC, University of New England , Armidale, NSW , Australia
Vonk Jennifer
Electronic publication date: 2019 Jun 7
Publication date: 2019
Volume: 7
Electronic Location ID: e7033
Received 2019 Feb 15; Accepted 2019 Apr 27
Copyright: © 2019 Monk et al.
Copyright year: 2019
Copyright holder: Monk et al.
License: This is an open access article distributed under the terms of the Creative Commons Attribution License, which permits unrestricted use, distribution, reproduction and adaptation in any medium and for any purpose provided that it is properly attributed. For attribution, the original author(s), title, publication source (PeerJ) and either DOI or URL of the article must be cited.
License URL: https://creativecommons.org/licenses/by/4.0/

Keywords: Cognitive bias, Vigilance, Animal welfare, Emotion, Stress-induced hyperthermia, Positive, Threat perception, Behavior, Livestock, Merino

Funding: Commonwealth Scientific and Industrial Research Organisation (CSIRO) University of New England (School of Environmental and Rural Science project expense support) This work was funded by the Commonwealth Scientific and Industrial Research Organisation (CSIRO) (internal funding) and the University of New England (School of Environmental and Rural Science project expense support). The funders had no role in study design, data collection and analysis, decision to publish, or preparation of the manuscript.

==============================
When an individual attends to certain types of information more than others, the behavior is termed an attention bias. The occurrence of attention biases in humans and animals can depend on their affective states. Based on evidence from the human literature and prior studies in sheep, we hypothesized that an attention bias test could discriminate between pharmacologically-induced positive and negative affective states in sheep. The test measured allocation of attention between a threat and a positive stimulus using key measures of looking time and vigilance. Eighty 7-year-old Merino ewes were allocated to one of four treatment groups; Anxious (m-chlorophenylpiperazine), Calm (diazepam), Happy (morphine) and Control (saline). Drugs were administered 30 min prior to attention bias testing. The test was conducted in a 4 × 4.2 m arena with high opaque walls. An approximately life-size photograph of a sheep was positioned on one wall of the arena (positive stimulus). A small window with a retractable opaque cover was positioned on the opposite wall, behind which a dog was standing quietly (threat). The dog was visible for 3 s after a single sheep entered the arena, then the window was covered and the dog was removed. Sheep then remained in the arena for 3 min while behaviors were recorded. Key behaviors included time looking toward the dog wall or photo wall, duration of vigilance behavior and latency to become non-vigilant. In contrast with our hypothesis, no significant differences were found between treatment groups for duration of vigilance or looking behaviors, although Anxious sheep tended to be more vigilant than Control animals (P < 0.1) and had a longer latency to become non-vigilant (P < 0.001). A total of 24 of 80 animals were vigilant for the entire test duration. This censoring of data may explain why no differences were detected between groups for vigilance duration. Overall, a lack of difference between groups may suggest the test cannot discriminate positive and negative states in sheep. We suggest that the censoring of vigilance duration data, the use of insufficient drug doses, the potential influence of background noise and the age of the sheep may explain a lack of difference between groups. Due to these potential effects, it remains unclear whether the attention bias test can detect positive states in sheep.

Introduction

The assessment of attention biases in non-human animals may allow us to gain a better understanding of the underlying affective states that relate to their welfare. An attention bias occurs when an animal attends to certain types of information before or for longer than others. Attention biases are determined by the salience of the information, or its perceived importance to the individual, and can be influenced by the animal’s transient emotional states and sustained by moods. For example, humans in anxious states pay more attention toward threatening information than non-anxious individuals (Bradley et al., 1995; Bradley, Mogg & Lee, 1997; Bar-Haim et al., 2007). Based on this principle, studies were conducted to determine whether an attention bias test developed for sheep was sensitive to changes in anxiety-like states (Lee et al., 2016; Monk et al., 2018b). The test measured time allocation of attention between a dog (threat) and a food reward (positive). The studies found that sheep in a pharmacologically-induced anxious state paid more attention toward the threat, evidenced by an increase in time spent looking toward the previous location of the dog and increased vigilance behavior compared to control animals. Further, sheep in an induced anxious state were less likely to feed than control animals. In comparison with controls, an induced calm state reduced time looking toward the threat, decreased vigilance and increased likeliness to feed. These results suggested the test could be used to assess anxious affective states in sheep. It remained unclear whether the pharmacological treatments modeled transient states or long-term moods, and whether the test could differentiate these types of states. Hereafter, we use the term affective states to encompass both transient emotions and moods.

The attention bias test for sheep was later modified by replacing the food with a photograph of a conspecific, to remove the potential influence of appetite on animal behavior (Monk et al., 2018a). In the modified method, sheep in pharmacologically-induced anxious and depressed states were more vigilant than control animals. However, the anxious group showed an attention bias toward the photograph rather than the threat. This unexpected result was attributed to the social aspect of the alternative positive stimulus and highlighted a need for context specific interpretations of behavioral responses. It appeared that within the context of the modified method, vigilance and exploratory behaviors, such as latency to sniff the environment, could be used to determine whether an animal was in a more negatively valenced affective state. The direction of attention could be used to discriminate anxious and depressed states from a neutral state. Although the interpretation of responses changed between test methods, each study demonstrated that an attention bias test could be used to assess and differentiate contrasting affective states in sheep, thus providing a new approach for researchers to better understand the affective states of livestock.

While many animal studies have focused on reducing the occurrence of negative affective states, the presence of positive affective states also comprises an important, but relatively understudied component of animal well-being (De Vere & Kuczaj, 2016). A number of human studies have demonstrated attention biases toward rewarding stimuli in subjects experiencing positive moods (Tamir & Robinson, 2007; Grafton, Ang & MacLeod, 2012; Sanchez & Vazquez, 2014; Caudek, Ceccarini & Sica, 2017), although these results have not been consistent in all tested populations (Isaacowitz et al., 2008). In sheep, Lee et al. (2016) and Monk et al. (2018b) demonstrated that the anxiolytic drug diazepam induced an attention bias away from a threatening stimulus and toward a food reward compared to saline treated control animals. This suggests that the attention bias test may be used to assess positive, or at least less negative, affective states in sheep. However, it could not be confirmed whether increased attention toward the food was due to the induced affective state or an increase in appetite caused by the drug (Foltin, 2004; Gaskins, Massey & Ziccardi, 2008). The influence of a non-negative affective state on behavior in the modified attention bias test has not yet been established.

Pharmacological agents can be useful for modelling different types of affective states in animals, in a more standardized and repeatable manner than many environmental manipulations (Mendl et al., 2009; Doyle et al., 2015). Further, drugs can be used that remain active for the duration of testing, and provide information on the mechanisms and pathways underpinning animal behavior. Few studies have used pharmacological treatments in sheep to induce and assess positive affective states. The anxiolytic drug diazepam was used to reduce anxiety-like behaviors during development of the original attention bias test method (Lee et al., 2016; Monk et al., 2018b). Further, it has been used to reduce anxiety-like behaviors in a range of other contexts, such as during isolation and suddenness tests (Drake, 2006; Destrez et al., 2012). It has also been shown to attenuate stress-induced hyperthermia in other species, consistent with a less anxious state (Olivier et al., 2003; Bouwknecht, Olivier & Paylor, 2007; Lee et al., 2017). However, for each of these studies it is difficult to determine whether the state induced by diazepam was truly positive as opposed to neutral or simply less negative than the other treatments. Verbeek et al. (2014) used the opioid agonist morphine to induce a positive affective state in sheep, due to its association with feelings of euphoria in humans (Riley et al., 2010). In sheep, the drug was found to enhance an optimistic judgement bias observed after feeding, consistent with having induced a more positive affective state (Verbeek et al., 2014). Therefore, morphine may be a useful pharmacological agent for confirming whether an attention bias test can discriminate euphoric-like positive affective states in sheep.

The aim of the current study was to determine whether the modified attention bias test could discriminate between positive and negative affective states in sheep. Specifically, we aimed to compare a high-anxiety state (Anxious), a low-anxiety state (Calm), a euphoric-like state (Happy) and controls, induced using the drugs m-chlorophenylpiperazine (mCPP), diazepam, morphine and saline, respectively. We hypothesized that the Anxious group would be more vigilant than control animals while the Calm and Happy groups would be less vigilant, in line with previous studies (Lee et al., 2016; Monk et al., 2018a, 2018b). Further, we hypothesized that the Anxious group would show an increased body temperature response to the injection and would spend more time looking toward the positive stimulus during testing (photograph of a conspecific), as shown by Monk et al. (2018a). Given the differences in interpretation of behavioral responses between the original and modified test methods, it was difficult to predict the direction of attention for the Calm and Happy groups. However, our preliminary hypothesis was that they would both also pay more attention toward the positive stimulus (photo), in line with the human literature. We did not have a priori hypotheses for how the Calm and Happy groups might differ from one another in vigilance or attention paid to the dog or photo. However, we did expect these groups would differ in behavioral measures of arousal such as activity and vocalizations, with the Happy group showing signs of a higher arousal state than the other groups, consistent with previous studies (Verbeek et al., 2012, 2014).

Materials and Methods

Animal ethics

The protocol and conduct of the experiments were approved by the CSIRO F.D. McMaster Laboratory Animal Ethics Committee (ARA18-17), under the New South Wales Animal Research Act 1985.

Experimental design

To test whether the attention bias test could differentiate between different types of affective states, drugs were used to pharmacologically induce contrasting positive and negative affective states in ewes prior to testing. Specifically, drugs were used to induce high-anxiety states (Anxious), low-anxiety states (Calm), euphoric-like states (Happy), and control states (Control).

A total of 80 Merino ewes were weighed and randomly distributed between the treatment groups balancing for bodyweight. Sheep then had numbers painted on their wool for individual identification and were divided into two cohorts (n = 40 per cohort) to be tested on separate days for logistical reasons. Treatment groups were evenly distributed between the two cohorts. All injections and tests for the 40 animals studied on a given day, occurred between 8.00 am and 1.30 pm. The experiment was conducted during October 2018.

Animal details

The sheep used in this experiment were non-lactating, non-pregnant Merino ewes, approximately 7 years old, with an average bodyweight of 47.0 ± 5.4 kg. Sheep were managed extensively throughout their life and were kept at pasture prior to testing. Sheep had prior experience with dogs during routine on-farm management when being moved between paddocks and handling facilities, but had no experience with the attention bias test. All sheep were bred, raised and tested on the same farm in Armidale, Australia.

Drug details

All drugs were administered as a single intramuscular (i.m.) injection into the rump of the animal, 30 min prior to testing in the attention bias test. An anxiety-like state was induced in the Anxious group using the anxiogenic drug mCPP (Tocris, Bristol, UK). This drug has previously been shown to significantly impact on animal behavior 30 mins after i.m. administration (Lee et al., 2016; Monk et al., 2018a, 2018b). The mCPP was administered at a dose rate of 1.5 mg/kg. This is a reduced rate compared to previous studies, as recommended by Monk et al. (2018a), due to the observation of abnormal behaviors at the higher dose rate of 2 mg/kg. Prior to treatment, mCPP was dissolved in BP Water for Injection (Baxter, Old Toongabbie, Australia) at a rate of 60 mg/ml. A calm-like state was induced in the Calm group using the anxiolytic drug diazepam (Troy Laboratories, Sydney, Australia). The diazepam was administered at a dose rate of 0.1 mg/kg as per previous studies (Lee et al., 2016; Monk et al., 2018b). The drug was administered i.m. rather than intra-venously (i.v.) for consistency across treatment groups. Studies in humans have shown complete bioavailability of diazepam after i.m. injection, with rapid absorption from muscle peaking at around 30 min (Divoll et al., 1983; Moore et al., 1991). A euphoric-like state was induced in the Happy group using the drug morphine (Hospira, Melbourne, Australia). The morphine was administered at a dose rate of 1 mg/kg as per previous studies (Verbeek et al., 2012, 2014). Plasma morphine concentration is shown to peak within 20 min after i.m. administration in humans, with 100% systemic availability (Stanski, Greenblatt & Lowenstein, 1978). In sheep, Morphine is shown to have a half-life of approximately 119 min in blood plasma after i.v. administration (Bengtsson et al., 2009). The Control group were given 1 ml of BP saline i.m. (Baxter, Old Toongabbie, Australia).

Attention bias test

The current study used the same attention bias test described by Monk et al. (2018a) (Fig. 1). Briefly, the test arena consisted of a concrete yard surrounded by 1.8 m high opaque walls. There was a small window located on one wall of the arena, which could be completely covered with a retractable opaque cover. Directly opposite the window was an approximately life-size photograph of an unfamiliar female conspecific. Once a sheep entered the arena, an unfamiliar dog was visible to the sheep through the window for 3 s, then the window was covered and the dog was removed. The sheep then remained in the arena for 3 min, while behaviors were recorded using a Sony Handicam video camera (model number HDR-XR550; Sony Electronics Inc., San Diego, CA, USA) that was positioned above the test arena in an adjacent building. A line of metal panels were positioned inside the arena so that sheep could not move into the corners out of view of the camera (Fig. 1). The total accessible area of the arena was 4 × 4.2 m.

Figure 1 Photograph of the attention bias test immediately after the test sheep entered the arena.

The dog was visible for 3 s, then a retractable opaque cover was lowered over the window and the dog was removed. Sheep remained in the test for 3 mins. The “#” symbol indicates the entrance of the arena, a camera was positioned above the arena to the right of the photograph (not visible in photograph). For a schematic diagram of the arena, see Monk et al. (2018a). Photo credit: Jessica Monk.

Behavioral measures in the attention bias test

The behaviors recorded in the attention bias test are summarized in Table 1. Most behaviors were collated from video footage using The Observer XT 12.0 (Noldus Information Technology, Wageningen, The Netherlands). Behaviors were continuously recorded for the test duration. During the video analysis the test arena was divided into nine grid sections (zones) which were overlaid on video footage, to calculate zones crossed, zones entered, and zone durations (Table 1). Open- and close-mouthed vocalizations were scored on the day of testing by an experienced hidden observer, who was positioned behind the opaque matting out of view of the sheep. The same observer also scored the dog’s behavior on a three point scale as: (1) quietly stood still, (2) lunged or crouched down, or (3) barked or growled at the sheep with any posture. A score of 3 was given on six occasions, for one animal in the Control group, two animals in the Anxious group, and three animals in the Calm group.

Table 1 Ethogram of behaviors recorded during the attention bias test (Monk et al., 2018b).

Behavior	Definition	
Attention	The direction in which the sheep is looking with binocular vision (Lee et al., 2016; Piggins & Phillips, 1996). The test arena was divided into four areas of attention: dog wall, photo wall, door wall, and back wall. Total duration of attention was recorded for the dog and photo walls. Duration looking at the other walls were not analyzed as these areas were not central to our hypotheses.	
Vigilance	Time spent with the head at or above shoulder height (Frid, 1997; Lee et al., 2016). Latency to become non-vigilant was also calculated.	
Sniff photo	Number of times and latency to sniff the photo.	
Sniff environment	Number of times and latency to sniff the floor or walls of the test arena.	
Vocalizations	Number of open-mouthed bleats and close-mouthed bleats were recorded separately.	
Zones crossed	Number of zones crossed with both front feet placed into the new zone, or one front foot in the zone and the other on the line.	
Zones entered	Number of zones entered (one to nine)	
Zone duration	Total time spent in each of the nine zones. Data for time spent in the zone closest to the photo were used for further analysis, as well as number of entries into the zone closest to the dog window.	
Urinations	Number of urinations.	

Sheep treated with mCPP were also monitored for abnormal behaviors previously described by Doyle et al. (2015). These included ataxic gait, tail shaking, head shaking, body shaking, and head rolling. Tail shaking was observed in four sheep during testing. No other abnormal behaviors were observed during testing.

Internal body temperature

Internal body temperature was recorded using Thermochron iButtons® (Model number DS1922L-F5, accuracy 0.5 °C, resolution 0.063 °C, weight 3.3 g; Embedded Data Systems, Lawrenceburg, KY, USA). The iButtons were attached to blank (progesterone-free) Controlled Internal Drug Release devices (CIDR®; Zoetis, Melbourne, Australia) using polyolefin heat-shrink tubing, as described by Lea et al. (2008). A CIDR was inserted into the vagina of each sheep one day prior to testing using an applicator lubricated with obstetrical lubricant. The iButtons were set to log at an interval of 20 s beginning 30 min prior to attention bias testing. Data were extracted using the program eTemperature version 8.32 (OnSolution, Castle Hill, Australia). Data from four temperature loggers were missing due to technical faults.

Body temperature data were extracted at times −30, −20, −10, −1, 6, 11, 15, 21, and 26 min relative to the beginning of attention bias testing. For each time point, the average of three consecutive temperature recordings were used. Times −1 and 6 min were identified as the average baseline and peak temperatures before and after attention bias testing across treatment groups.

Statistical methods

Data were analyzed using R version 3.5.1 (R Development Core Team, 2018). P-values less than 0.05 were considered significant and P < 0.1 were considered a tendency toward significance. All model residuals were checked for normality and homoscedasticity using Shapiro–Wilks test for normality and visual assessment of Q–Q and residuals vs fitted values plots. Treatment group, test order within each test day and dog behavior score were fitted as fixed effects in all linear models. Test order and dog behavior did not reach significance in any of the models and were subsequently removed using a backward elimination approach, considering both the Akaike and Bayesion information criterion. Cohort (test day) was fitted as a random effect in all mixed effects models. The package nlme was used to fit linear mixed effects models (Pinheiro et al., 2016). The package lme4 was used to fit generalized linear mixed effects models (Bates et al., 2015). Post hoc multiple comparisons were conducted using a Tukey method for adjustment of P-values. Where significant differences were found between groups, effect sizes were estimated using Pearson’s correlation coefficient r (Field, Miles & Field, 2012). Estimates of effect size were not made for count data.

Attention and vigilance data were analyzed using linear mixed effects models. Data for attention to photo wall and vigilance were log transformed to meet normality assumptions. A Kruskal–Wallis non-parametric ANOVA was also used to confirm the analysis of the vigilance data as the residuals were only marginally improved by transformation. Post hoc multiple comparison tests for the Kruskal–Wallis ANOVA were performed using the package pgirmess (Giraudoux, 2018).

Photo sniff frequency and number of zones entered were analyzed using generalized linear mixed effects models with a Poisson distribution for count data. Sniff environment frequency and zones crossed were analyzed using generalized linear mixed effects models with a negative binomial distribution due to evidence of over-dispersion. Vocalization data were analyzed using a negative binomial hurdle model using the package pscl, to account for the presence of excess zeros in the dataset (Zeileis, Kleiber & Jackman, 2008). Data for time spent in the zone closest to the photo were analyzed using a linear mixed effects model. The number of animals in each group that entered the zone closest to the dog wall were analyzed using a Fisher’s Exact Test. Post hoc multiple comparisons between groups were performed using the package rcompanion (Mangiafico, 2018). Urination data were also analyzed using a Fisher’s Exact Test, examining the number of animals in each group that urinated.

All latency data were analyzed with Cox’s proportional hazards model using survival analysis (Therneau & Grambsch, 2000; Therneau, 2015), as described by Monk et al. (2018b). These data included latencies to sniff the photo, sniff the environment and become non-vigilant. Animals that failed to perform each behavior within 180 s were deemed as censored results. Treatment and test day were fitted as fixed effects in all proportional hazards models.

All body temperature data were analyzed using a linear mixed effects model, fitting treatment, time, and a treatment × time interaction as fixed effects. Sheep identity nested within test day was fitted as a random effect to account for the repeated measurements across time points. A subset of the body temperature data was taken from time −1 min onward to better assess the influence of attention bias testing itself on body temperature responses. Change in body temperature from time −1 min was analyzed in the same way as the total temperature dataset, fitting a linear mixed effects model to account for repeated measures over time.

Results

Attention and vigilance

Raw attention and vigilance duration data are summarized in Fig. 2. Linear mixed effects models showed that duration of attention toward the dog and photo walls did not differ significantly between treatment groups (Table 2; Fig. 3). However, the Anxious and Calm groups tended to spend less time looking toward the dog wall than the Control and Happy groups (P < 0.1).

Figure 2 Boxplots displaying the distribution of observed duration data.

Boxplots show the median durations, the interquartile range (IQR) and the range of data within 1.5 × the IQR for duration of attention to the dog wall (A), attention to the photo wall (B), vigilance (C), and time spent standing in the zone closest to the photo (D). The dots represent raw duration data for each individual sheep within the treatment groups. We note that the plot axes are scaled differently to more clearly display the data within each observed variable.

Table 2 Mean ± s.e.m. behavioral responses of sheep in each treatment group during the attention bias test.

Behavioral measure	Anxious	Control	Calm	Happy	Test method	Test value (df)	P-value	
Attention to dog wall (s)	53.3 ± 6.7	64.9 ± 6.7	52.3 ± 6.7	63.6 ± 6.7	LME	F(3,76) = 2.4	0.075	
Attention to photo wall (s)1	3.7 ± 0.2 (38.9)	3.5 ± 0.2 (33.4)	3.8 ± 0.2 (44.8)	3.5 ± 0.2 (33.6)	LME	F(3,76) = 2.0	0.123	
Vigilance (s)1,2	5.1 ± 0.1 (175.6)	5.1 ± 0.1 (169.7)	5.1 ± 0.1 (170.5)	5.1 ± 0.1 (171.6)	LME	F(3,76) = 0.6	0.30	
Vigilance (mean rank)	52.5 ± 5.0	33.2 ± 4.5	36.5 ± 5.3	39.9 ± 4.9	Kruskal–Wallis	X2(3) = 8.1	0.04	
Sniff photo (n)1	1.0 ± 0.1 (2.7)a	1.4 ± 0.1 (4.0)a,b	1.6 ± 0.1 (4.7)b	1.1 ± 0.1 (3.1)a	GLME_P	X2(3) = 14.2	0.016	
Sniff environment (n)1	0.0 ± 0.4 (1.0)a	1.1 ± 0.4 (2.9)a,b	1.3 ± 0.4 (3.6)b	1.0 ± 0.5 (2.7)a,b	GLME_NB	X2(3) = 9.9	0.02	
Sniff closed window (n)3	1	1	1	4	FET	N/A	0.34	
Zones crossed (n)1	2.7 ± 0.1 (14.1)a	2.9 ± 0.1 (17.9)a	2.8 ± 0.1 (16.5)a	3.5 ± 0.1 (32.7)b	GLME_NB	X2(3) = 22	<0.001	
Zones entered (n)1	1.5 ± 0.1 (4.5)a	1.7 ± 0.1 (5.1)a	1.7 ± 0.1 (5.5)a,b	2.0 ± 0.1 (7.5)b	GLME_P	X2(3) = 17.2	<0.001	
Standing near photo (s)	102.8 ± 16.5a,b	105.0 ± 16.5a	98.2 ± 16.5a,b	70.4 ± 16.5b	LME	F(3,76) = 3.3	0.025	
Enter zone close to dog (n)3	4a	6a,b	7a,b	14b	FET	N/A	0.009	
Open-mouthed bleats (n)	0.7 ± 0.5a	1.5 ± 0.6a	3.3 ± 1.2a	16.5 ± 2.6b	GLME_NB_H	X2(3) = 11.5	0.009	
Close-mouthed bleats (n)	2.8 ± 0.6	4.1 ± 0.6	6.0 ± 0.9	7.4 ± 0.7	GLME_NB_H	X2(3) = 4.0	0.26	
Urinations (n)3	7	3	2	4	FET	N/A	0.31	
Notes:

a,b Different superscripts within rows indicate a significant difference between treatments as determined using post hoc analyses, significant P-values are emphasized with bold font.

1 Least squares means are given on the log scale, back-transformed means are given in parentheses.

2 Vigilance duration was censored at 180 s.

3 Raw number of animals of a total of 20 in each group are given.

LME, linear mixed effects model fitting test day as a random effect; GLME, generalized linear model with a Poisson (P) or Negative Binomial (NB) distribution; data including excess zeros used hurdle models (H); FET, Fisher’s Exact Test was used to calculate probability; test statistic is not applicable (N/A); post hoc analyses were performed using the package rcompanions.

Figure 3 Mean ± s.e.m. time spent looking toward the dog wall and the photo wall for each treatment group during attention bias testing.

The times spent looking toward the dog and photo walls were analyzed using linear mixed effects models, fitting treatment as a fixed effect and test day as a random effect.

The linear mixed effects model on vigilance duration data showed no significant differences between treatment groups (Table 2). The Kruskal–Wallis ANOVA on vigilance duration data showed an overall treatment effect; however, post hoc multiple comparisons showed no differences between the groups (Table 2). Survival analyses showed the Anxious group had a significantly higher latency to become non-vigilant than the other groups, while the Calm and Happy groups tended to have a higher latency to become non-vigilant than the Control group (Table 3; Fig. 4).

Table 3 Hazard ratios for latency to sniff the photo, sniff the environment, and become non-vigilant as affected by treatment group.

Latency to	Group	Mean (s)1	Censored (n)2	Coefficient3	SE (coeff)	Hazard ratio4	Wald (z)	P	Likelihood ratio	df	P	
Sniff photo	Control	19.1a	1	Reference					18.4	3	0.001	
Anxious	65.7b	4	−1.188	0.36	0.31 (0.15–0.62)	−3.3	0.001	
Calm	16.4a	0	−0.095	0.33	0.91 (0.48–1.73)	−0.29	0.772	
Happy	18.9a	0	−0.299	0.33	0.74 (0.39–1.43)	−0.89	0.372	
Anxious			Reference					
Calm			1.093	0.35	2.98 (1.49–5.97)	3.09	0.002	
Happy			0.889	0.35	2.43 (1.23–4.81)	2.56	0.010	
Calm			Reference					
Happy			−0.204	0.32	0.816 (0.44–1.53)	−0.63	0.526	
Sniff environment	Control	102.4a	5	Reference					20.5	3	<0.001	
Anxious	147.3b	12	−1.189	0.44	0.3 (0.12–0.72)	−2.69	0.007	
Calm	102.5a	8	−0.256	0.39	0.77 (0.36–1.65)	−0.66	0.510	
Happy	91.2a	4	0.128	0.36	1.13 (0.56–2.3)	0.35	0.723	
Anxious			Reference					
Calm			0.934	0.46	2.54 (1.03–6.27)	2.03	0.043	
Happy			1.317	0.44	3.73 (1.58–8.82)	3	0.003	
Calm			Reference					
Happy			0.384	0.38	1.46 (0.69–3.11)	1	0.317	
Non-vigilance	Control	65.5a	2	Reference					19.3	3	<0.001	
Anxious	132.9b	11	−1.522	0.42	0.21 (0.09–0.49)	−3.65	0.000	
Calm	99.0a	5	−0.671	0.35	0.51 (0.25–1.02)	−1.9	0.058	
Happy	100.2a	6	−0.674	0.36	0.51 (0.25–1.02)	−1.88	0.060	
Anxious			Reference					
Calm			0.851	0.42	2.34 (1.02–5.37)	2.01	0.044	
Happy			0.848	0.43	2.33 (1–5.43)	1.97	0.049	
Calm			Reference					
Happy			−0.003	0.37	0.99 (0.48–2.06)	−0.01	0.993	
Notes:

1 Raw mean latencies are given, superscripts indicate significant differences between groups for each behavior, significant P-values are emphasized with bold font.

2 Number of animals which failed to exhibit the given behavior within 180 s were deemed as censored results.

3 Regression coefficient from the Cox-proportional hazards model.

4 95% confidence interval given in parentheses.

Figure 4 Kaplan–Meier curves for latency to sniff the photo (A), sniff the environment (B), and become non-vigilant (C).

Each time an animal exhibited the given behavior, the probability on the Y-axis drops.

Other behaviors

The Calm group had the highest frequencies of sniffing the photo and environment while the Anxious group sniffed the photo and environment the least; however, neither group differed significantly from the Controls (Table 2). Anxious sheep had a longer latency to sniff the photo and environment than all other treatment groups (Table 3; Fig. 4).

The Happy group crossed and entered more zones than the other groups, while the other groups did not differ (Table 2). The Happy sheep also spent the least amount of time standing near the photo (r = 0.26–0.30) and performed more open-mouthed vocalizations than the other groups (Table 2; Fig. 2). More sheep in the Happy group entered the zone closest to the dog wall compared to the Control, Calm and Anxious groups (Table 2). No statistical differences were found between groups for the number of animals that urinated (Table 2).

Body temperature

The repeated measures analysis on body temperature data from the baseline at −30 min showed a significant Treatment × Time interaction (F(24, 576) = 4.5, P < 0.001). Contrasts from the model summary indicated that body temperature did not differ between groups at times −30 or −20 min (P > 0.1) (Fig. 5). At time −10 min, the Anxious group had a significantly higher body temperature than the Calm and Happy groups, but only tended to be higher than the Control group (t (71) = −1.8, P = 0.069, r = 0.21). The Anxious group had a higher body temperature than the other groups at all other time points (Fig. 5, r = 0.26–0.34). The Control, Calm and Happy groups did not differ at any time point.

Figure 5 Mean ± s.e.m. body temperatures for the Anxious (●), Control (■), Happy (x), and Calm (▲) groups.

All injections were administered at time −30. The gray bar denotes the time of attention bias testing. The letters (a, b) indicate significant differences between groups at time −10. The “*” symbol denotes a significant difference between the Anxious group mean and all other groups as determined using a repeated measures linear mixed model.

For change in body temperature after time −1 min, the Time × Treatment interaction was not significant (F(15, 360) = 0.70, P = 0.78). The model fitting fixed effects only without the interaction showed no significant effect of treatment on change in body temperature (F(3, 72) = 0.24, P = 0.865).

Discussion

The pharmacological treatments induced a number of significant effects on behaviors and body temperature; however, in contrast with our hypothesis, the key behaviors of vigilance and looking time did not differ between treatment groups in the current study. There are a number of possible explanations for this, which will be discussed in greater detail throughout the following paragraphs. Firstly, these results could suggest that the modified attention bias test cannot provide a reliable measure of affective states in sheep. Treatment differences between the Control and Anxious groups observed previously during a modified attention bias test were not replicated in the current study (Monk et al., 2018a). This could mean that the results observed in the previous study were an anomaly. It is worth noting, however, that the Anxious group tended to spend less time looking toward the dog wall and displayed a higher level of vigilance than the Control animals, albeit not strongly supported statistically, which is consistent with our expectations and the previous study (Monk et al., 2018a). Further, the Anxious group did have a significantly longer latency to become non-vigilant than the other groups. Finally, the Calm group tended to spend less time looking toward the dog wall than the Control and Happy groups. Thus, we would suggest the current study does not indicate the attention bias test is not useful as a measure of affective states in sheep, but rather that it may not be as sensitive to these changes as previously shown, or that additional factors impacted on animal behavior during this study, as discussed below.

One external factor that may have influenced animal behavior in the current study was the presence of background noise during testing. Throughout both test days, there was unexpected work being conducted in a nearby sheep handling facility. This meant there was some distant background noise consisting of conspecific vocalizations and vehicle movement. At the time of testing, the researchers had not expected this level of noise to impact on the behavior of the sheep being tested. Further, the noise was consistent and repetitive, spanning across the test period so that all treatment groups were exposed to a similar level of noise. However, the noise may have caused the test animals to reach a maximum level of vigilance. The overall time spent vigilant was higher in the current study relative to previous studies (Lee et al., 2016; Monk et al., 2018a, 2018b). Further, there was a high number of censored data points, with 24 of 80 animals showing vigilance behavior for the entire test duration. The prevalence of censored data was much higher than in the previous study, during which only 4 of 50 animals were vigilant for the entire duration of the test (raw data generated by Monk et al., 2018a). If the duration of vigilance reached a behavioral and temporal maximum within the given test duration, this would have reduced the amount of variation in the data and power of the test to discriminate between treatment groups. Notably, more sheep in the Anxious group were vigilant for the entire test duration. Thus, for vigilance duration, the Anxious group mean was impacted to a greater extent than other groups, and may have disproportionately disguised the extent of heightened vigilance in this group. This is reflected by the survival analysis on latency to become non-vigilant, which accounted for the censoring of vigilance data and showed a significant treatment effect. The background noise may have also impacted on looking time measures, as the noise was coming from the direction of the dog wall. However, the mean times spent looking toward the dog wall for all animals were 59 and 61 s for the current study and Monk et al. (2018a), respectively, suggesting the influence on direction of attention was minimal. Nevertheless, it cannot be ruled out that the presence of background noise impacted on vigilance and looking behaviors in the test. Consideration of the potential influence of background noise on animal responses will be of particular importance if applying the test in a commercial, on-farm setting, in which affective states are induced by the animals’ environments.

An alternative explanation for a lack of response could be that the drugs or doses were inappropriate for modifying affective states or did not induce the expected affective states. During the current study, the Calm animals tended to spend less time looking toward the dog wall as predicted. However, differences between the Calm and Control groups were not significant for any observed behavior, contrasting with previous studies using the same dose rate of diazepam (Lee et al., 2016; Monk et al., 2018b). During the original attention bias test method, vigilance, and eating behaviors were mutually exclusive. This means that increased time spent eating would have automatically caused a reduction in vigilance behavior. This is important to note as diazepam has been shown to increase appetite in other animal species (Foltin, 2004; Gaskins, Massey & Ziccardi, 2008). Thus, the decrease in vigilance shown previously may have been due to a drug effect on appetite rather than the influence of a calm state on attention biases in sheep. Another factor worth noting is that the drug had previously been administered i.v. rather than intra-muscularly. In humans, complete bioavailability of diazepam has been demonstrated after intra-muscular injection, with peak levels occurring around 30 min (Divoll et al., 1983; Moore et al., 1991). However, no studies in sheep have administered diazepam intra-muscularly at the same dose rate used in the current study. As diazepam or its metabolites were not directly measured in the current study, the efficacy of the drug cannot be confirmed. Further, injection of diazepam did not attenuate the stress-induced hyperthermia caused by attention bias testing. This contrasts with previous studies in cattle (Lee et al., 2017) and rodents and suggests the drug may not have had a strong anxiolytic effect in the current study (Olivier et al., 2003; Bouwknecht, Olivier & Paylor, 2007). It is also worth noting that a number of studies using diazepam in livestock and humans have found inconclusive or inconsistent effects (Clarke, Trim & Hall, 2014; Doyle et al., 2015; Lee et al., 2017; Mandelli, Tognoni & Garattini, 1978). Further studies including pharmacokinetic assessment are required to validate the use of diazepam as an anxiolytic treatment in sheep and other livestock species.

Evidence for the use of morphine to manipulate affective states in sheep is currently limited, with few studies using the drug with this intended effect. In the current study, morphine had no effect on key measures of attention bias, but did seem to increase arousal as predicted. Happy sheep crossed and entered more zones and were more vocal than Control animals, which is consistent with previous studies using the same dose of morphine (Verbeek et al., 2012, 2014). While it appears morphine has an effect on arousal, its influence on the valence component of affect is less clear. Happy sheep did not show reduced vigilance or increased exploratory behavior during the test, previously thought to indicate a less-negative state (Monk et al., 2018a). Instead, increased activity during isolation can be considered a fearful response for sheep (Romeyer & Bouissou, 1992; Forkman et al., 2007). Previously, Verbeek et al. (2012) demonstrated changes in ear posture after morphine treatment which were suggestive of decreased fearfulness (Reefmann et al., 2009). Verbeek et al. (2014) demonstrated an enhanced optimistic judgement bias after consumption of a food reward, suggestive of a more positive mood after treatment (Paul, Harding & Mendl, 2005). Therefore, it is possible that increased activity during the current study did not reflect increased fearfulness, but rather supports the suggestion by Verbeek et al. (2012) that the opioid system may be involved in the arousal component of affective state. The potential influence of morphine on the valence component of affect in sheep remains inconclusive. Further studies validating the use of this drug, or exploring alternative methods for inducing positive affective states, would be useful for understanding affective states and attention biases in sheep.

The drug mCPP has been used previously to induce anxious states in sheep, including studies of attention bias. However, the current study used a reduced rate of 1.5 mg/kg compared to the rate of 2 mg/kg used previously (Lee et al., 2016; Monk et al., 2018a, 2018b). It could therefore be possible the reduced dose-rate was insufficient to cause an equivalent anxiety-like response. Although mCPP did not potentiate the stress-induced hyperthermia caused by attention bias testing, the drug itself caused a significant increase in body temperature compared to all other treatment groups, which is consistent with an anxiety-like state and previous studies (Sherwood, Klandorf & Yancey, 2005; Bouwknecht, Olivier & Paylor, 2007; Pedernera-Romano et al., 2010; Lee et al., 2017; Monk et al., 2018a). The Anxious group also showed behavioral signs of increased anxiety during testing, displaying a higher latency to become non-vigilant and a higher latency to sniff the photo and environment compared to Control animals (Romeyer & Bouissou, 1992; Beausoleil, Stafford & Mellor, 2005; Beausoleil et al., 2012). Doyle et al. (2015) had also found that a lower dose of 1 mg/kg induced anxiety-like behaviors in young sheep during runway, startle and isolation tests. Overall, it appears that the dose of 1.5 mg/kg did cause an anxious-like response prior to and during testing. It is recommended that further studies continue to use the reduced dose rate of 1.5 mg/kg in adult sheep to lessen the presence of unwanted side-effects. We suggest the key behaviors of vigilance duration and looking time did not differ between the Anxious and Control groups due to the censoring of vigilance data or the influence of other confounding factors.

Additional factors such as animal age, sex, parity, and experience with dogs may have also influenced animal responses in the test. Previously, attention biases have been demonstrated in both male and female sheep ranging from 5 months old to 2 years old, while the current study assessed 7-year-old ewes. In humans, age has been found to alter the direction of attention biases (Isaacowitz et al., 2008). In sheep, previous experience with humans, age and parity have been shown to alter fearfulness (Viérin & Bouissou, 2002; Dodd et al., 2012). Notably, routine handling procedures on-farm, such as shearing and mustering to handling facilities with the use of dogs, are associated with indicators of behavioral and physiological stress in sheep and are shown to influence animal behavior during future procedures (Dwyer, 2009). Given that these older animals predominantly had negative experiences with humans and dogs throughout their lives, they may have developed a negative association with handling, which may have influenced their responses during testing. Additionally, animal age may have impacted the effect and metabolism of the drugs used in the study. For example, in humans, age is shown to impact the volume distribution and clearance rate of diazepam, as well as the metabolite concentration and peak time (Klotz et al., 1975; Divoll et al., 1983). In rats, morphine is shown to have a variable effect on behavior and nociception in 8-week-old vs 24-week-old animals (Paul, Gueven & Dietis, 2018). Understanding the potential influence of these factors on animal responses will allow for a clearer interpretation of behavior during attention bias studies.

Another factor that should be noted is the choice of positive and negative stimuli in the current test methodology. The negative stimulus was a live dog, which was presented for only 3 s of the test. The positive stimulus was a photograph of a conspecific, which was present for the entire test duration. As such, the positive and negative stimuli used in the current study were not balanced in intensity and presentation time. Within the context of the current study, attention biases are being assessed within treatment groups relative to other treatment groups. This means the key question is not whether an individual pays more attention toward the dog window relative to the photograph, but rather whether a treatment group pays more attention toward the dog window relative to the other treatment groups being tested. Consequently, discrepancies between the positive and negative stimulus intensities and stimulus presentation times should not have impacted on our results, as these factors were consistent across all tested animals. However, it cannot be ruled out that the drugs may have impacted an animal’s perception of the stimuli and therefore their responses. For example, if the Anxious sheep perceived the photograph as a conspecific to a greater degree than the other groups, this could explain an additional attraction to the stimulus. Further studies should be conducted to better understand how animals perceive and respond to the stimuli presented during attention bias tests. Such studies should not only consider the salience of the stimuli, but should also consider the type of stimuli used with regards to the primary sensory systems of the animals being tested (Raoult & Gygax, 2018, Winters, Dubuc & Higham, 2015).

Additionally, it is important to consider that the stimuli used in the current study may not be generalizable to other studies and populations. We used a single dog for all tests to control for the potential effects of dog disposition or temperament on sheep responses. However, different dogs may evoke different behavioral responses in sheep during future studies. We note here that the photograph used in the current study has been made publicly available by Monk et al. (2018a) for use in further research; however, this photo may be less suitable for studies in different sheep breeds. Sheep have been shown to discriminate valenced photographs of conspecifics, and to generalize this discrimination of valenced faces to photos of different individuals (Bellegarde et al., 2017). As such, we do not expect that minor changes to conspecific photographs would have a great impact on animal responses. Nevertheless, future studies aiming to better understand how animals perceive and appraise different types of stimuli should take this into consideration.

The current study highlights a number of key areas for further research. Studies understanding the potential influence of factors such as age, experience and background noise on animal responses may allow the method to be adjusted or interpreted accordingly for different populations of animals or for different testing environments. It is suggested the test might be better suited to younger groups of animals that have had less experience with dogs and with routine handling practices that could influence their responses in the test. Inclusion of a habituation period could reduce the overall vigilance levels by reducing the fear-eliciting elements of the test itself, allowing clearer separation of vigilance between animals (Erhard, Elston & Davidson, 2006). However, this would limit the test’s application to a larger population of animals in an on-farm setting. Understanding the influence of background noise will be of particular importance if applying the test in a commercial, on-farm setting. It would be useful to examine the impact of other environmental and pharmacological manipulations on responses in the modified attention bias test. Studies examining environmental manipulations should utilize experiences and environments that are relevant to livestock production systems to facilitate the future application of the test in an on-farm setting. Studies examining pharmacological models should aim to better understand the way in which the models impact on the valence of affective state. This will best be done by utilizing a variety of methods, such as place preference tests and operant conditioning tasks. Finally, further studies should be conducted to examine the influence of different types of positive and negative stimuli on animal responses.

Conclusions

It remains unclear whether the attention bias test can be used to detect positive affective states in sheep. Further, the current study was not able to replicate previous findings that negative affective states influenced responses in the modified attention bias test. It is suggested the current study should be repeated on a population of younger animals, making sure to reduce or eliminate background noise during testing, which may have confounded results. Further studies should be conducted to confirm the effects of the given pharmacological agents and to ensure the doses and administration routes are appropriate to induce specific affective states. It may be useful to explore alternative pharmacological agents for inducing affective states in sheep, and to examine the impact of environmental manipulations on attention biases.

The authors would like to thank the staff and students at CSIRO for their assistance during the experiments: Jim Lea, Tim Dyall, Troy Kalinowski, and Koli the dog. We would like to thank Alison Small for her advice and assistance with the drug treatment protocol. The first author would like to thank the Sheep CRC for supporting her postgraduate studies.

Additional Information and Declarations

Competing Interests

Author Contributions

Animal Ethics

Data Availability

The authors declare that they have no competing interests.

Jessica E. Monk conceived and designed the experiments, performed the experiments, analyzed the data, prepared figures and/or tables, authored or reviewed drafts of the paper, approved the final draft.

Caroline Lee conceived and designed the experiments, performed the experiments, authored or reviewed drafts of the paper, approved the final draft, attained funding for the project.

Sue Belson conceived and designed the experiments, performed the experiments, approved the final draft.

Ian G. Colditz conceived and designed the experiments, authored or reviewed drafts of the paper, approved the final draft.

Dana L.M. Campbell conceived and designed the experiments, performed the experiments, authored or reviewed drafts of the paper, approved the final draft.

The following information was supplied relating to ethical approvals (i.e., approving body and any reference numbers):

The protocol and conduct of the experiment was approved by the CSIRO F.D. McMaster Laboratory Animal Ethics Committee, Armidale, Australia. The project was approved (approval number ARA18-17) under the New South Wales Animal Research Act 1985.

The following information was supplied regarding data availability:

The dataset generated for this article can be found in the CSIRO data access portal (DAP): Monk, Jessica; Lee, Caroline; Belson, Sue; Colditz, Ian; Campbell, Dana (2019): AEC18/17 Assessing positive affective states in sheep using attention bias. v1. CSIRO. Data Collection. DOI 10.25919/5c6269bc895ed

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
