# Peer review of "The influence of pharmacologically-induced affective states on attention bias in sheep"

_PeerJ, doi:10.7717/peerj.7033_

## Round 0.1 · original submission · Minor Revisions

I think your paper is well-written and the methods are carefully thought out. I think that the study makes a valuable contribution to the literature despite the fact that the results are unclear. Although your discussion is balanced, I think the findings warrant even greater skepticism regarding aspects of the procedure. Both reviewers have given very helpful and thorough advice, which I agree with. I don't think that the requested revisions constitute a major revision so I hope you will be willing to undertake the work to make the clarifications that they ask for. I do not have any additional comments at this point except that I think the issue of using a dog as a threatening stimulus needs to be more carefully considered, especially given the subjects' familiarity with dogs. Can you justify this choice? Obviously, it would be better to have avoided the confound of live and photographic stimuli of different types. At the very least, you'd have a condition where the sheep was live and an image of a dog was presented as a control. One other comment is that there is a lot of literature on judgement bias using procedures with ambiguous stimuli that should also at least be mentioned.

·

Basic reporting

Dear Jessica, dear Caroline, dear co-authors,

As you may guess my identity quite easily anyway (and the editors did not think that our past and current exchanges are too much of a conflict of interest), I make this a non-anonymous review.

As with your other recent work, I very much appreciate your approach, which is presented in clear language, refers to the relevant literature and is properly structured. In respect to the formalities, I am unsure only because it seems that raw data should be supplied which, at least, was not included in the materials available to me.

In my view, a balanced set of stimuli is necessary for an attention bias test such that, in a neutral state (your control group), both stimuli elicit a similar amount of attention. Only then, we can hope to see changes towards either the negative or the positive stimulus according to some experimental manipulation. I assume that the “memory” of the dog (that is no longer visible) and the picture of a real sheep is an attempt to achieve such a balance. Perhaps, you could be more explicit about this? Moreover, the implication of the fact that the stimuli to which attention is directed are different (live animal versus photograph; memory versus current presence) would strengthen the discussion in my opinion. Given Fig. 2, the balance in respect to attention in the control group was more or less though not ideally achieved (they directed more attention towards the dog-wall than the sheep-wall). Does that have consequences on what you try to find

Your conclusion mainly focuses on positive states (the inclusion of which was novel in this experiment; e.g. line 41). As far as I can see, you could not replicate the previous results (introduced on lines 54-49 and 67-73) as clearly, either. I think this merits some more space in the discussion. I think the notion that the previous experiments could have been spurious results needs to be considered as well. Even if, in the end, the arguments that you list and that indicate that the current experiment may have had its problem are the most likely explanation for the lack of differences in your current study.

Quite often, you talk about emotional “states” without specifying whether you mean short-term emotions or longer-term mood. I think, at least conceptually, this makes quite a difference because, e.g. emotions are much more specific than mood states. I think the manuscript could be improved if you were more specific in this respect or explain the complications about being more specific and what your results could imply for these different types of emotional states.

It seems that you have a clear notion (which is well argued for) what states you consider more or less negative, “neutral”, and positive. I think it would help the reader if you would list the different treatments in this order (from strongly negative to positive) throughout in the text and in the figures. One might expect some monotonous trend (or some [inverse-]u-shape) across these treatments and the existence of such a relationship would be much more easily visible if the treatments were accordingly sorted.

As far as I can see the internal body temperature is not really introduced. Therefore, it seems to come as a little surprise on lines 197ff. As you provide this data at some length, a more proper introduction with specific hypothesis may be useful.

Experimental design

As far as I can see, the experiment was well designed and executed. I have some minor to moderate comments on your statistical procedures, though, that you might want to reconsider in a revised version of the manuscript.

Overall, I find it quite hard to follow which statistical procedures were used for which outcome variable. Perhaps, this could be added to one of the tables. Moreover, in some instances, it is not clear to me, which fixed and random effects were included in which model. Some more detailed listing would be here as well in my view. In addition, it may be useful to know which specific procedures from which packages you used for the different models. A detail: if you used R 3.5.1 it should be cited as R Core Team 2018 (and not 2015).

I would very much appreciate if there was one single table for all the different outcomes that were evaluated. This provides an easy overview and the possibility for an assessment of multiple testing issues (the increasing probability for finding a random low p-value with each additional test conducted). This would mean to combine Table 2 (in which global tests are given) and Table 3 (only pair-wise comparisons, why no global test?) and move those evaluations only mentioned in the text to the table, too (lines 276-284).

In my view, a bias can best be explained as the concurrent focus of attention. Therefore, the first outcome variable that I would be interested in would be the proportion of time spent attentive on the dog wall divided by the total attention spent on the dog or sheep wall. (The overall attention to these walls could be used as a weight in the evaluation to account for the fact that this proportion can more exactly be calculated the longer the attentive period). This would directly show a shift from one wall to the other. It would also account for the fact that the sheep in the test could not be attentive to both walls at the same time (sum-one-constraint). I am not sure whether and how this pattern could be seen directly in the absolute time that was directed to one or the other wall. Similarly, I miss the amount of time standing near (or the relative proportion of dog-door versus sheep photo) or sniffing the dog door (or, again, the relative proportion).

I was unsure whether all of the outcome variables are really needed. E.g. is there really an expectation in respect to the attention to the door wall, the back wall or sniffing the environment? If yes, I think these would need to be stated. I would rather tend to drop these outcome variables.

Moreover, I think some outcome variables are evaluated several times in slightly different ways, e.g. the duration of vigilance, failure to be non-vigilant, and vigilant for 180 sec are basically the same outcome variable. In my view, the survival analysis does the proper job for censored durations here and should be the only model reported for vigilance. Similar cases exist for the categorised outcomes that are only mentioned in the text and for the other two survival models (sniffing photo and sniffing environment).

Some more statistical details:
• Line 218: how was test order coded? Test order across treatments or within a test-day? You did not include any interactions? Was there a specific reason for this?
• Line 220: Type of elimination approach? Based on which criterion?
• Line 230: How can a Fisher-Test be applied to a contingency table larger than 2 x 2?
• Lines 242ff: Did you include the same random effect (clustering) as in the mixed models? If not, why not?
• Lines 247: Usually, such auto-correlations are only valid for equally-spaced time points. This is not the case here. Did you somehow adjust for that?
• Lines 287ff: Main effects can only be interpreted in a model including an interaction if sum-contrasts are used. Did you do so?
• Lines 296-297: I am lost here. I am not sure what you did nor why this (subset?) was evaluated.
• Fig. 3: Obviously, many survival analyses use the terminology of “failure-time”. Here, I find this term confusing. Could you use another Y-axes description?
• Fig. 4: This data was not right-skewed, either? I am not sure what the full and what the post-hoc models were here (specifically for the comparisons within a time-point, there is no repeated measurements of the individuals, right?). In my view, the random effect here might have included the sheep identity nested in the test day. Was that the case?

In respect to statements as on lines 215-216, I advise the reading of the ASA statement on the use of p-values (https://doi.org/10.1080/00031305.2016.1154108), but overall this has no major implication here. But e.g. in Table 2, the close-mouthed bleats reach a large p-value only even though the mean differences between the groups are quite large.

Validity of the findings

I always admired your pharmacological approach such that I was almost comforted a little that this approach does not automatically lead to a straightforward interpretation of results. You discuss this issue well and at length. I was wondering whether there are some data of e.g. place preference tests with sheep receiving one or the other drug that you could cite or whether such tests could be recommended as possible next steps to solve some of the issues that you raise. Alternatively, do you have other recommendations how the effect of the different drugs should be evaluated in sheep?

As stated above, I think a somewhat more critical general discussion (lack of replication of previous results) would be good.

Additional comments

I have only some minor additional comments:
• The abstract may need some revising in respect to the results if some of the issues mentioned are changed.
• Line 46: only before? It could also be for longer within a certain time window, right?
• Lines 119-120: Could you elaborate a bit more why this was your expectation?
• Line 163: I am aware that you previously define what you mean, e.g. by “happy”. Nevertheless, this is one place where just using “happy” struck me as a bit too simplistic (and anthropocentric). Could you re-evaluate your terminology in this respect? E.g. using the drug name instead of the presumed state?
• Line 175: Could this process of removal be perceived (heard) by the sheep? Do you have any (qualitative) indication that the sheep knew whether the dog was still there or not?
• Lines 188-190: Should/could you state which treatments were involved for the 6 occasions of score 3?
• Results: As you summarise some of the results in the text, I am often unsure which specific outcome variables you address. Could you be more specific in the text?
• Lines 342ff: But you did find difference between the treatments. Therefore, this ceiling effect would not have been a worry for all of the treatments, right? Also, the fact that censoring took place mainly in one of your groups, actually makes the groups more distinct and should therefore not be a problem (as it is accounted for in the survival analysis).
• Line 396: “less clear” and then the difference between happy and other sheep: is this not a contradiction?
• Figures: I would welcome if some more of the outcome variables were shown graphically. Also, I very much like to see some of the raw data as well (e.g. by using boxplots).
• Table 2: Could you include the degrees of freedom in this table, too?
• Conclusion: I would like to see a mention of the “negative” treatments as well.
• Line 447: Are dose-response studies the only necessary studies or could alternative approaches be useful (see above)?

I hope to see a revised version of your manuscript soon and I think given my comments above, this should not be too hard to achieve. Best wishes, Lorenz (Gygax, HU Berlin)

Reviewer 2 ·

Basic reporting

The literature is appropriate and well referenced, except for R version 3.5.1 (it is more probably the R Core Team 2018 than 2015 and the full reference should be “R Core Team. 2018. R: A language and environment for statistical computing. R Foundation for Statistical Computing, Vienna, Austria. URL http://www.R-project.org/”. Everything else is fine.

Experimental design

No comment, but see (minor) comments on the Materials & Methods section.

Validity of the findings

No comment, but I would have liked to see more discussion about future studies going towards a more on-farm setting of the attention bias test, and critics about the stimuli used (see my comments).

Additional comments

The authors present a modified attention bias test to assess emotionally induced positive and negative states in sheep. The approach is interesting and overall well explained. The study is well performed and in general well reported but see my detailed suggestions thereafter:

Abstract
Lines 20 and 63: add a dash between “pharmacologically” and “induced”.
Lines 23-24 and 35: for consistency with the manuscript put a capital letter for each treatment group: “Eighty 7-year-old Merino ewes were allocated to one of four treatment groups: Anxious (m-chlorophenylpiperazine), Calm (diazepam), Happy (morphine) and Control (saline)” and “Anxious group”.
Lines 25, 30, 153, 163, 204, 208 and 378: use the SI units, i.e. “min” instead of “mins”
Line 25: You say there that the test arena was “4 x 4 m” while line 179 it is “4 x 4.2 m”, so please be consistent.

Introduction
Lines 45-46: please precise that “An attention bias occurs when an animal attends to certain types of information before or for longer than others”, as you explain it after.
Lines 46-48: could you please reformulate this sentence that is currently unclear.
Line 51: “principle” instead of “principal”
Line 56 and 64: I suggest reformulating “Anxious sheep” to “anxious-induced sheep” as you use the formulations anxiety-like states, induced calm and controls in this paragraph referring to previous studies.
Lines 79-80: include the “e.g.” in the references’ parenthesis.
Lines 90-92: I understand that pharmacological agents can be used in a more standardized and repeatable manner that environmental manipulations, however I think you should discuss in the discussion section) the advantage of using various environmental manipulations to induce emotional states for future applications in more on-farm settings. Animals that are not experimental animals have their emotional states influenced by the environment, and not by drugs.
Line 98: add “e.g.” in the parenthesis, and if possible give examples of other contexts where diazepam was used to reduce anxiety-like behaviours.
Line 105: remove the coma after euphoric-like.
Line 118: add that the Calm and Happy groups “would both also pay more attention […]”.
Line 122: replace “morphine treated group” by “Happy group”.

Materials & Methods
Lines 142-143: could you please rephrase. It is weird to say at what time the first injection was (without specifying that it was always 30 min prior to testing) and when the last test was performed.
Lines 147-149: could you precise whether the sheep were used to be handle? I know the sheep were 7 years old but they were only handle for (negative) handling procedures it could have an impact on how they reacted to the test, e.g. being highly vigilant. Also, please precise whether the sheep were bred, raised and tested on the same farm and how they were housed, specifically before the attention bias test.
Line 151: can you provide the peak effect of all the drugs used: it is always 30 min after i.m. injection?
Line 153: remove the capital letter of “Anxious” state.
Line 161: define “i.v.” used for the first time in the manuscript.
Lines 164-166: precise the mode of administration. Was it also i.m. for the morphine and saline?
Line 169: in Figure 1, please precise in the legend that two video cameras were used: a first camera was position to obtain the view angle presented in this picture while a second camera was positioned above the arena to the right of the photograph.
Line 172: “life-size” instead of “life-sized”
Line 173: precise that the photograph was from an unfamiliar female conspecific. Also, you should discuss the fact you used the same dog and photograph of sheep for all the test. This can be seen as problematic for generalisation of the test because you observe responses to a specific dog/photograph. Additionally, I would like to see discussed the difference between the two stimuli used: one is a live animal whereas the other one is a life-size 2D colour photograph. Concerning the experimental set-up, after 3 s you hide the dog by covering the window: don’t you think that this movement might attract vigilance towards the window?
Lines 176-177: add the detail about the camera right after you talk about it such as “[…] behaviors were recorded using a Sony Handicam video camera (Sony, Australia, model number HDR-XR550) positioned above the test arena in an adjacent building”.
Line 182: in Table 1, could you precise whether you assess the positions of the sheep ears to define its vigilance behaviour. A sheep with passive ears for instance cannot be defined as vigilant. Also, did you measure the attention and vigilance behaviours continuously, or e.g. every 1 s? Moreover, see my comments below on the elimination definition (as you present the urination behaviour only).
Lines 184-188: explain the hypotheses behind the behavioural measures (i.e. activity, arousal)
Line 199: remove the double parenthesis next to each other such as “(Model number DS1922L-F5, accuracy 0.5°C, resolution 0.063°C, weight 3.3g; Embedded Data Systems, Lawrenceburg, USA)”.
Line 206: It seems, according to the data files, that data from 4 temperature loggers were missing (sheep No. 20/40/41/70). Please correct it or explain the missing data.
Line 215: according to the R version used, it is more probably the R Core Team 2018 than 2015. Also, please provide the full reference detail: R Core Team. 2018. R: A language and environment for statistical computing. R Foundation for Statistical Computing, Vienna, Austria. URL http://www.R-project.org/.
Lines 236-238: give the reference about the pscl package right after you talk about it such as “Vocalization data were analyzed using a negative binomial hurdle model using the package pscl (Zeileis, Kleiber & Jackman, 2008), to account for the presence of excess zeros in the dataset”.
Line 239: you analysed the eliminations that is according to your definition the number of urinations and defecations, however you give the data and present the results only for urinations. I guess there were not enough defecations for the analysis, but please precise it or change the elimination definition.
Lines 242-243: give the reference of the analysis right after you talk about it such as “All latency data were analyzed with Cox’s proportional hazards model using survival analysis (Therneau & Grambsch, 2000; Therneau, 2015), as described by Monk et al. (2018b)”.

Results
Line 266: “Control group” instead of “control group”
Lines 276-278: please precise that it is on 20 sheep for each group and add the missing comas such as “More sheep in the Happy group entered the zone closest to the dog wall (14/20) compared to the Control, Calm and Anxious groups (6/20, 7/20 and 4/20 sheep, respectively; Fishers Exact Test, P=0.009)”.
Lines 280-281: I suggest changing the sentence to “Seven, 4, 3 and 2 sheep from the Anxious, Happy, Control and Calm groups, respectively, urinated during testing”. Also, see above my comments about sheep’s defecation behaviour.
Line 286: This paragraph, with all its statistical results, is difficult to read. Maybe you could make a table to synthesize them. Be consistent in the way you present the P-values (P in capital letter and no space before and after the equal sign).
Line 289: add the missing “=” in “(X2 (8) = 870.6, P<0.001)”
Line 291: add “min” such as “at times -30 or -20 min (P>0.1) (Fig 4). At time -10 min, […]”. In the Figure 4’s title replace “Mean (± se)” with “Mean ± s.e.m.”

Discussion
Lines 307-308: I suggest to replace “displayed a higher, albeit not significantly higher” with “displayed a higher, albeit not strongly supported statistically”
Line 308: add a capital letter to “Control animals”
Line 317: are you sure you want to introduce the background noise during test as the first explanation for your results, though I agree the background noise is important when testing attention and you clearly and sincerely discuss the issue.
Line 323: add a coma after “However”
Line 326: “there was a high number of […]” instead of “there were a high number of [...]”
Line 329: remove the double parenthesis such as “(unpublished data from Monk et al., 2018a)”
Line 337: put “respectively” between comas
Line 341: Here would be the good place to discuss future application of the attention bias test in more on-farm settings, in which the emotional states are induced by the animals’ environment. Also, I would like to see somewhere in the discussion some critics about the stimuli used (live vs. 2D image and no variability in the stimuli) and therefore how generalizable the approach is.
Lines 348, 352 and 360: precise “anxious-like” response/state
Lines 381-384: please rephrase the sentence in two sentences. Also, position the references after introducing the studies such as “previous studies in cattle (Lee et al., 2017) and rodents (Olivier et al., 2003; Bouwknecht, Olivier & Paylor, 2007) […]”.
Line 391: remove “in sheep” that is redundant in this sentence.
Lines 432-433: I understand that lengthening the test duration could potentially overcome the high number of censored data points observed for vigilance behaviour, however I would not recommend this because then test sheep might not be focus for a long enough time and you take the risk of not being able to explain why they (not) react in such a way.

---

## Round 0.2 · Minor Revisions

Thank you for a MS that required very little editing. I am happy to formally accept the MS as soon as the following very minor issues are attended to:

In a few places, “which” should be replaced with “that” (e.g., line 261, 269, 306, 498, 507)
Place a , after However on line 284.
Line 289 (and similarly on lines 297, 327, 486), place a ; after effect and a , after however.
Place commas around however on line 334, 502.
You should report estimates of effect size where appropriate.
On line 453, “on” is not necessary.
Figure 3 seems to show an interaction although I cannot easily see where that was tested in the Results. The fact that there is a bigger difference between treatments for control and happy sheep makes me wonder if it is more the case that the pharmacological agents were not effect for anxious and calm groups rather than that the testing paradigm is ineffective, although I realize this is just one of many patterns of results. You do mention possible issues with dose in the discussion but I would be careful not to focus overly on questioning the paradigm without additional validations of the effectiveness of the dose.

---

## Round 0.3 · accepted · Accept

Thank you for your response to these last few queries. I am happy to formally accept this interesting manuscript.

#